# STOCHASTIC SUBSET SELECTION FOR EFFICIENT TRAINING AND INFERENCE OF NEURAL NETWORKS

## ABSTRACT

Current machine learning algorithms are designed to work with huge volumes of high dimensional data such as images. However, these algorithms are being increasingly deployed to resource constrained systems such as mobile devices and embedded systems. Even in cases where large computing infrastructure is available, the size of each data instance, as well as datasets, can provide a huge bottleneck in data transfer across communication channels. Also, there is a huge incentive both in energy and monetary terms in reducing both the computational and memory requirements of these algorithms. For non-parametric models that require to leverage the stored training data at the inference time, the increased cost in memory and computation could be even more problematic. In this work, we aim to reduce the volume of data these algorithms must process through an end-to-end two-stage neural subset selection model, where the first stage selects a set of candidate points using a conditionally independent Bernoulli mask followed by an iterative coreset selection via a conditional Categorical distribution. The subset selection model is trained by meta-learning with a distribution of sets. We validate our method on set reconstruction and classification tasks with feature selection as well as the selection of representative samples from a given dataset, on which our method outperforms relevant baselines. We also show in our experiments that our method enhances scalability of non-parametric models such as Neural Processes.

## 1 INTRODUCTION

The recent success of deep learning algorithms partly owes to the availability of huge volume of data (Deng et al., 2009; Krizhevsky et al., 2009; Liu et al., 2015), which enables training of very large deep neural networks. However, the high dimensionality of each data instance and the large size of datasets makes it difficult, especially for resource-limited devices (Chan et al., 2018; Li et al., 2019; Bhatia et al., 2019), to store and transfer the dataset, or perform on-device learning with the data. This problem becomes more problematic for non-parametric models such as Neural Processes (Hensel, 1973; Kim et al., 2019a) which require the training dataset to be stored for inference. Therefore, it is appealing to reduce the size of the dataset, both at the instance (Dovrat et al., 2019; Li et al., 2018b;b) and the dataset level, such that we selects only a small number of samples from the dataset, each of which contains only few selected input features (e.g. pixels). Then, we could use the selected subset for the reconstruction of the entire set (either each instance or the entire dataset) or for a prediction task, such as classification.

The simplest way to obtain such a subset is random sampling, but it is highly sub-optimal in that it treats all elements in the set equally. However, the pixels from each image and examples from each dataset will have varying degree of importance (Katharopoulos & Fleuret, 2018) to a target task, whether it is reconstruction or prediction, and thus random sampling will generally incur large loss of accuracy for the target task. There exist some work on coreset construction (Huggins et al., 2016; Campbell & Broderick, 2018; 2019) which proposed to construct a small subset with the most important samples for Bayesian posterior inference. However, these methods cannot be applied straightforwardly to deep learning with an arbitrary target task. How can we then sample elements from the given set to construct a subset, such that it suffers from minimal accuracy loss on any target task? To this end, we propose to learn a sampler that *learns* to sample the most important samples for a given task, by training it jointly with the target task and additionally meta-learn a sampler over a distribution of datasets for instance selection in the classification task.

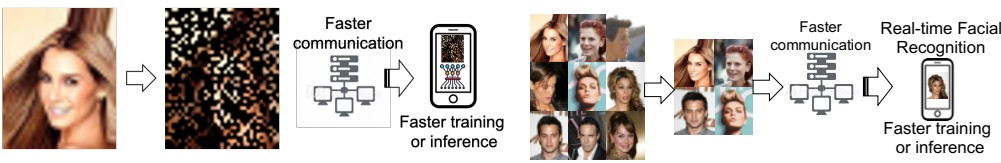

(a) Features Selection           (b) Instance Selection

Figure 1: **Concept**: Our Stochastic Subset Selection method is generic and can be applied to many type of sets. **(a)** Selecting a subset of features (pixels) out of an image. This reduces communication cost in data transfer between devices and allows faster training or inference on resource-constrained devices due to the reduced computational cost. **(b)** Selecting a subset of instances out of a dataset. This helps resource-constrained system to train faster, or to make non-parametric inference more scalable.

Specifically, we learn the sampling rate for individual samples in two stages. First we learn a Bernoulli sampling rate for individual sample to efficiently screen out less important elements. Then, to select the most important elements out of this candidate set considering relative importance, we use a Categorical distribution to model the conditional distribution of sampling each element given a set of selected elements. After learning the sampling probability for each stage, we could perform stochastic selection of a given set, with linear time complexity. Our *Stochastic Subset Selection (SSS)* is a general framework to sample elements from a set, and it can be applied to both *feature sampling* and *instance sampling*. SSS can reduce the memory and computation cost required to process data while retaining performance on downstream tasks.

Our model can benefit from a wide range of practical applications. For example, when sending an image to an edge device with low computing power, instead of sending the entire image, we could send a subset of pixels with their coordinates, which will reduce both communication and inference cost. Similarly, edge devices may need to perform inference on a huge amount of data that could be represented as a set (e.g. video, point clouds) in real-time, and our feature selection could be used to speed up the inference. Moreover, our model could also help with on-device learning on personal data (e.g. photos), as it can select out examples to train the model at a reduced cost. Finally, it can help with the scalability of non-parametric models which requires storage of training examples, such as Neural Processes, to scale up to large-scale problems.

We validate our SSS model on multiple datasets for 1D function regression and 2D image reconstruction and classification for both feature selection and instance selection. The results show that our method is able to select samples with minimal decrease on the target task accuracy, largely outperforming random or an existing sampling method. Our contribution in this work is threefold:

- We propose a novel two-stage stochastic subset selection method that **learns to sample** a subset from a larger set with linear time complexity, with minimal loss of accuracy at the downstream task.

- We propose a framework that trains the subset selection model via **meta-learning**, such that it can generalize to unseen tasks.

- We validate the efficacy and generality of our model on various datasets for **feature selection** from an instance and **instance selection** from a dataset, on which it significantly outperforms relevant baselines.

## 2 RELATED WORK

**Set encoding - Permutation invariant networks**  Recently, extensive research efforts have been made in the area of set representation learning with the goal of obtaining order-invariant (or equivariant) and size-invariant representations. Many propose simple methods to obtain set representations by applying non-linear transformations to each element before a pooling layer (e.g. average pooling or max pooling) (Ravanbakhsh et al., 2016; Qi et al., 2017b; Zaheer et al., 2017; Sannai et al., 2019). However, these models are known to have limited expressive power and sometimes not capable of capturing high moments of distributions. Yet approaches such as Stochastic Deep Network (De Bie et al., 2018) and Set Transformer (Lee et al., 2018) consider the pairwise (or higher order) interactions among set elements and hence can capture more complex statistics of the distributions . These methods often result in higher performance in classification/regression tasks; however, they have run time complexities of $O(n^2)$ or higher.

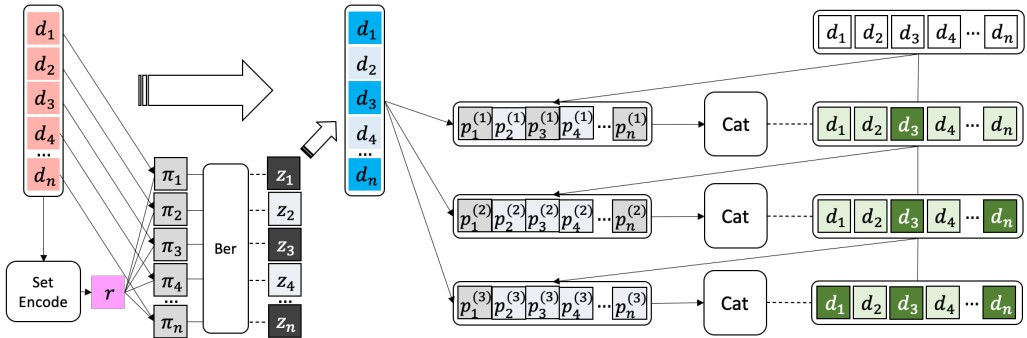

(a) Candidate Selection with Bernoulli distribution    (b) Iterative Coreset Selection with a stack of Categorial distributions

Figure 2: **Overview.** The subset $D_s$ is sampled by a stochastic 2-stage subset selection process. Dark shaded boxes correspond to selected elements while lightly shaded boxes correspond to non-selected elements in the given set. Also, Ber and Cat correspond to the Bernoulli and Categorical distributions respectively. **(a)** Candidate(in blue) Selection. **(b)** Core Subset(in green) Selection using continuous relaxations of discrete distributions.

**Subset sampling**   There exist some works which have been proposed to handle large sets. Dovrat et al. (2019) proposed to learn to sample a subset from a set by generating $k$ virtual points, then matching them back to a subset of the original set. However, such element generation and matching process is highly inefficient. Our method on the other hand only learns to select from the original elements and does not suffer from such overhead. Wang et al. (2018) proposed to distill the knowledge of a large dataset to a small number of artificial data instances. However, these artificial data instances are only for faster training and doesn't capture the statistics of the original set. Moreover, the instances are generated artificially and can differ from the original set making the method less applicable to other tasks. Also several works (Qi et al., 2017a;c; Li et al., 2018b; Eldar et al., 1997; Moenning & Dodgson, 2003) propose *farthest point sampling*, which selects $k$ points from a set by ensuring that the selected samples are far from each other on a given metric space.

**Image Compression**   Due to the huge demand for image and video transfer over the internet, a number of works have attempted to compress images with minimal distortion. These models (Toderici et al., 2017; Rippel & Bourdev, 2017; Mentzer et al., 2018; Li et al., 2018a) typically consist of a pair of encoder and decoder, where the encoder will transfer the image into a compact matrix to reduce the memory footprint and communication cost, while the decoder is used to reconstruct the image back. These methods, while achieving huge successes in the image compression problem, are less flexible than ours. Firstly, our model can be applied to any type of sets (and instances represented as sets), while the aforementioned models mainly work for images represented in tensor form. Furthermore, our method can be applied both at the instance and dataset level.

**Representation learning**   Our instance-sampling model is also related to the Variational Auto Encoder (VAE) (Kingma & Welling, 2013). However, while VAE learns a compact representation of a *data point*, our model learns a compact representation of a *set*. Balın et al. (2019) learns a global feature selection model for reconstruction of the input data from selected features via unsupervised learning. Chen et al. (2018) learns instancewise feature selection with the goal of model interpretation by extracting subset of features most informative for a given sample. Our method also falls in this category.

**Active Learning**   Active learning methods are aimed at selection of data points for labeling given a small labelled set. This domain is different from our method since active learning does not consider the label information but our method does utilize label information. Also, our motivation is quite different. We focus on efficiency in inference and training of non-parametric models by reducing the sizes of the inputs, be it pixels or instances and this greatly differs from the goal of active learning. Methods such as (Sener & Savarese, 2017; Coleman et al., 2019; Wei et al., 2015) all tackle the data selection problem in the active learning setting.

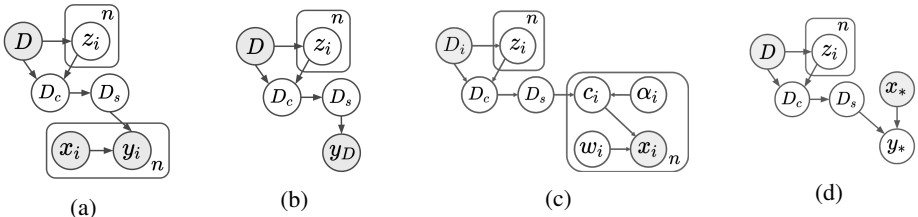

Figure 3: **Graphical Models: (a)** Feature Selection for Reconstruction. **(b)** Feature Selection for Prediction Task. **(c)** Instance Selection model. **(d)** Instance Selection model for Classification

# 3 APPROACH

## 3.1 PRELIMINARIES

In this work, we consider data of the type $D = \{d_1, \ldots, d_n\}$ where individual $d_i$'s are possibly represented as *input* $x_i$ and *target* $y_i$. $D$ is the complete set and we assume that within $D$, there exists a subset $D_s = \{s_i, \ldots, s_k\} \subset D$ such that $k \ll n$ and that for an arbitrarily defined loss function $\ell(., D)$ that we are interested in optimizing over the full set $D$, $D_s$ can be used as a proxy for $D$ such that $\ell(., D) \approx \ell(., D_s)$. In what follows, we present a method that learns the conditional distribution $p(D_s|D)$ of the subset $D_s$ via a two stage selection procedure dubbed *candidate selection* and *autoregressive subset selection*. The overall objective is then to minimize the loss function with respect to the subset $D_s$, $\mathbb{E}_{p(D_s|D)}[\ell(., D_s)]$. When the set $D$ itself follows a distribution of sets as in the *meta-learning* framework, then the objective becomes $\mathbb{E}_D[\mathbb{E}_{p(D_s|D)}[\ell(., D_s)]]$. In essence, we seek to construct a subset $D_s$ that is optimally representative of the full set $D$ w.r.t $\ell(.)$.

## 3.2 STOCHASTIC SUBSET SELECTION

In order to select $D_s$, we need to model the interactions among the elements of $D$ and construct $D_s$ based on said interactions. However, when the cardinality $|D|$ of the set $D$ is large or it's elements $d_i$'s are high dimensional, modeling such pairwise interactions becomes computationally infeasible. As such, we first present the *candidate selection* procedure used to construct a smaller set, $D_c$, without considering inter-sample dependencies. This is then followed by the *autoregressive subset selection* procedure used to construct $D_s$ from $D_c$ by modeling inter-sample dependencies. The complete model is depicted in Figure 2.

## 3.3 CANDIDATE SELECTION

We model the task of candidate selection as a random Bernoulli process where the logits of the Bernoulli function are conditioned on the set representation of the full set $D$ and the individual elements $d_i \in D$. For a set $D$ with cardinality $n$ we define $Z := \{z_i\}_{i=1}^n$ such that $z_i \in \{0, 1\}$ and $z_i = 1$ implies that $d_i \in D_c$ and for each $d_i$, $z_i$ is computed according to:

$$p(z_i|d_i, D) = \text{Ber}(z_i; \rho(d_i, r(D))), \tag{1}$$

where $r(D)$ is a permutation-invariant function that compresses $D$ into a single vector set representation and $\rho(d_i, r(D))$ computes the logits used to calculate the probability of $d_i$ belonging to $D_c$. We implement both $r(D)$ and $\rho(d_i, r(D))$ as neural networks and specifically for $r(D)$, we use Deep Sets (Zaheer et al., 2017). Since *Ber* is non-differentiable, we use the continuous relaxations of the Bernoulli distribution introduced in (Maddison et al., 2016; Jang et al., 2016; Gal et al., 2017).

Specifically, to sample $z_i$, we execute the following computational routine:

$$z_i = \sigma\Big(\frac{1}{\tau}\Big(\log\frac{\pi_i}{1-\pi_i} + \log\frac{u}{1-u}\Big)\Big), \quad \pi_i = \rho(d_i, r(D)), \quad u \sim \text{Unif}(0, 1), \tag{2}$$

where $\sigma$ is the Sigmoid function, $\tau$ is the temperature for the continuous relaxation and $u$ is sampled from the uniform distribution. $\tau$ is set to 0.05 in all our experiments. Given that pair-wise interactions

---

**Algorithm 1** Fixed Size Subset Selection

---

  **Input**      $k$(subset size), $q$(# elements selected at each iteration), $D = \{d_1, d_2, \ldots, d_n\}$ (Full Set)
  **Output**    $D_s = \{s_1, s_2, \ldots, s_k\}$ (selected subset)
1:  **procedure** STOCHASTIC SUBSET SELECTION($k, q, D$)
2:      $(\pi_1, \pi_2, \ldots, \pi_n) \leftarrow (\rho(d_1, r(D)), \ldots, \rho(d_n, r(D)))$
3:      $z_i \sim \text{Ber}(\pi_i)$ for $i = 1, \ldots, n$.
4:      $D_c \leftarrow \{d_i \text{ for } i = 1 : n \text{ if } z_i\}$                               $\triangleright$ Candidate Selection
5:      $D_s \leftarrow \emptyset$
6:      **for** $i = 1, \ldots, k/q$ **do**                         $\triangleright$ AutoRegressive Subset Selection
7:          $D_s \leftarrow D_s \cup \text{AUTOSELECT}(q, D_s, D_c)$             $\triangleright$ Select $q$ elements
8:      **return** $D_s$
9:  **procedure** AUTOSELECT($q, D_s, D_c$)
10:     $C = \{w_1, w_2, \ldots, w_m\} \leftarrow D_c \setminus D_s$
11:     $(p_1, p_2, \ldots, p_m) \leftarrow (f(w_1, D_c, D_s), f(w_2, D_c, D_s), \ldots, f(w_m, D_c, D_s))$
12:     $(p_1, p_2, \ldots, p_m) \leftarrow (p_1, p_2, \ldots, p_m) / \sum_{j=1}^{m} p_j$
13:     $Q \leftarrow$ Select $q$ elements from $C$ with probability $(p_1, p_2, \ldots, p_m)$
14:     **return** $Q$

---

between elements are not considered in this stage, learning $p(z_i|d_i, D)$ ensures that highly activating samples are selected instead of a random subset of the original set.

### 3.4 AUTOREGRESSIVE SUBSET SELECTION

The candidate selection stage can introduce samples with redundant information in $D_c$ since no effort was made to compare the informativity of the elements. To alleviate this issue, we must first model the interactions between the elements of $D_c$ and construct $D_s$ based on the relative importance of individual elements. To construct a representative subset $D_s$ with $|D_s| = k$, $k$ iterative steps are required and at step $i$ the probability of an element in $D_c \setminus D_s^{(i-1)}$ belonging to $D_s$ is computed according to:

$$p(s_i = d | D_c, D_s^{(i-1)}) = \frac{f(d, D_c, D_s^{(i-1)})}{\sum_{d' \in D_c \setminus D_s^{(i-1)}} f(d', D_c, D_s^{(i-1)})} \quad \forall d \in D_c \setminus D_s^{(i-1)}, \qquad (3)$$

where $D_s^{(i-1)}$ is the constructed subset at iteration $i-1$ and $f$ is a positive function. The key to avoiding samples with redundant information in $D_s^{(k)}$ lies in the fact that for each element added to $D_s$, it's selection is conditioned on both $D_c$ and all elements in $D_s^{(i-1)}$. We further propose a method that samples $q$ elements from $\text{Cat}(p_1, \ldots, p_m)$ in a single pass for efficient training. Specifically, instead of sampling $q$ times from the categorical distribution, we can sample the selection mask for element $j$ from $\text{Ber}(q * p_j)$. In this routine, the probability of the element $j$ being selected is $q * p_j$ which is very close to the original distribution. Algorithm 1 details the entire procedure. The inference complexity depends heavily on the choice of the function $f$. If $f$ considers the pairwise interactions between all candidate elements and the selected elements, the inference complexity is $O(n) + O(k^2 d/q)$ where $n, d, k$ correspond to $|D|, |D_c|$ and $|D_s|$ respectively. In our experiments, for the choice of the function $f$, we utilize either a Set Transformer(Lee et al., 2018) or DeepSets(Zaheer et al., 2017) to model the pairwise interactions between the elements of a given set.

### 3.5 CONSTRAINING THE SIZE OF $D_c$

For computational efficiency, we may desire to restrict the size of $D_c$ to save computational cost when constructing $D_s$. We adopt the idea of Information Bottleneck and constrain the distribution of $Z$ for $D_c$. Specifically,

$$\mathbb{E}_{p(D)}[\mathbb{E}_{p(D_s|D)}[\ell(., D_s)] + \beta \text{KL}[p(Z|D)||r(Z)]] \qquad (4)$$

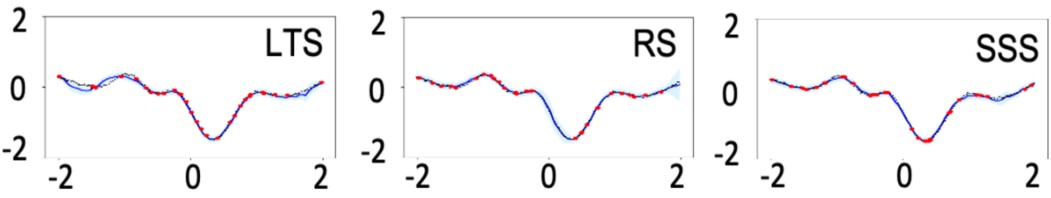

Figure 4: Models' performance on set reconstruction and classification task (lighter-color areas present the standard deviations). **(a)** 1D Function reconstruction. **(b)** CelebA (reconstruction). **(c)** CelebA (classification)

Figure 5: 1D Function Reconstruction.

where $r(Z)$ is a sparse prior. In our experiments, we set the parameter of the Bernoulli spares prior $r(Z)$ to either $0.1$ or $0.01$ for different levels of sparsity and $\beta$ is set to $1.0$ or $0.001$.

### 3.6 TASKS

We now present four tasks for which the described subset selection method is applied.

**Set Reconstruction** Given $D_s$ and a network $p_\theta(Y|X, D_s)$ parameterized by $\theta$, the task is to reconstruct $D = (X, Y)$. The objective function for this task is given as:

$$\mathbb{E}_{p(D)}[\mathbb{E}_{p(D_s|D)}[-\log p_\theta(Y|X, D_s)] + \beta \mathrm{KL}[p(Z|D)||r(Z)]] \quad (5)$$

Minimizing this objective ensures that we learn a compact subset($D_s$) most representative of $D$ and $D_s$ can then be used for other tasks. We implement $p_\theta(Y|X, D_s)$ as an Attentive Neural Process (ANP) (Kim et al., 2019b). An ANP takes as input a context($D_s$ in this case) and predicts a distribution of the elements in the original set $D$. It mimics the behaviour of a Gaussian Process but with reduced inference complexity. The complete model is depicted in Figure 3a.Experimental results for this task can be found in Section 4.1.

**Set Classification/Prediction** We can also opt to train the network to predict a single target $y_D$ for the set $D$. For instance, the target could be the class of an image(classification) or the statistics of the set(regression problem). Here, $p_\theta(y_D|D_s)$ is a neural network that predicts the target $y_D$. A set in this task, may be the features from a single example like an image and experimental results can be found in Section 4.2.The model for this task is depicted in Figure 3b. The objective function for this task is given as:

$$\mathbb{E}_{p(D)}[\mathbb{E}_{p(D_s|D)}[-\log p_\theta(y_D|D_s)] + \beta \mathrm{KL}[p(Z|D)||r(Z)]] \quad (6)$$

**Dataset Distillation: Instance Selection** For this task, we are given a dataset $\mathcal{D} = \{D_1, \ldots, D_n\}$ where each $D_i$ is a set of data points sampled from the entire dataset. Using CelebA as in illustrative example, some $D_i$ may consist of $|D_i|$ randomly sampled faces from the whole dataset. The goal is to construct $D_s$ for each $D_i \in \mathcal{D}$. We describe a model capable of taking as input $D_i \in \mathcal{D}$ to perform a task such as the reconstruction of all elements in the given dataset.

| Model | # Pixels | Storage | mAUC |
|---|---|---|---|
| Full Image | All 38804 | 114KB | 0.9157 |
| RS | 500 | 5KB | 0.8471 |
| SSS(rec) | 500 | 5KB | 0.8921 |
| SSS(MC) | 500 | 5*5KB | 0.9132 |
| SSS(ours) | 500 | 5KB | 0.9093 |

Table 1: CelebA Attributes Classification.

For a single dataset $D_i \in \mathcal{D}$, we apply the subset construction method already described to a $D_s$ that can be used to reconstruct all the elements in $D_i$. In essence, $D_i$ is *distilled* into a new dataset $D_s$ with $k < |D_i|$ elements. The task then is to reconstruct the entire set $D_i$ back conditioned only

| #Instances | 2 | 5 | 10 | 15 | 20 | 30 |
|---|---|---|---|---|---|---|
| FPS | 6.50 | 4.51 | 3.07 | 2.75 | 2.71 | 2.29 |
| Random | 3.73 | 1.16 | 0.90 | 0.38 | 0.39 | 0.20 |
| SSS(ours) | **2.53** | **1.02** | **0.59** | **0.33** | **0.24** | **0.17** |

| #Instances | 1 | 2 | 5 | 10 |
|---|---|---|---|---|
| FPS | 0.432 | 0.501 | 0.598 | 0.636 |
| Random | 0.444 | 0.525 | 0.618 | 0.663 |
| SSS(ours) | **0.475** | **0.545** | **0.625** | **0.664** |

Table 2: . FID Score for varying Instance Selection

Table 3: Accuracy on *mini*Imagenet

on $D_s$. As a first step, we represent $D_s$ as a unique representative vector $c$ for each element in the dataset akin to the statistics network used in the Neural Statistician (Edwards & Storkey, 2016) model. Specifically, to reconstruct an element $d_i \in D_i$ given $D_s$, $c$ is computed by applying a stochastic cross-attention

mechanism on $D_s$ where the stochasticity is supplied by a query $\alpha$ which is computed using $d_i$. To obtain varying styles in the generated images, we additionally learn a latent variable $w$ used to perturb $c$ and both are combined to obtain a new element $x$. The graphical model for this process is depicted in Figure 3c. Additionally, to ensure that $c$ is properly learnt, we add an informativity loss by reconstructing $c$ from the generated samples from the given dataset. The objective for the model depicted in Figure 3c for a single dataset $D$ is :

$$
\mathcal{L}(\theta, \phi, \psi) = \sum_{d_i \in D} [\mathbb{E}_{q_\phi(w_i|d_i)}[p_\theta(d_i|w_i, c_i)] - \mathrm{KL}[q_\phi(w_i|d_i)||p_\psi(w)]
$$
$$
- \mathrm{KL}[q_\phi(\alpha_i|d_i)||p_\psi(\alpha)] - \mathrm{KL}[q_\phi(c_i|D_s, \alpha_i)||p_\phi(c)]]
$$
(7)

where $p_\psi(\cdot)$ are priors on their respective latent variables and $q_\phi(\cdot)$'s are implemented with neural networks. All priors are chosen to be Gaussian with zero mean and unit variance. This objective is combined with the informativity loss on all samples in $D_i$. It is important to note that $c$ is computed using only $D_i$ for every element in $D$. In addition to Equation 7 and the informativity loss, the model is optimized together with the subset selection model already described. When the model is fully optimized, it is applied to the instance selection task on the given dataset. In summary, the purpose of the generative model introduced is to train the subset selection module for the instance selection task. Experimetal results for this task can be found in Section 4.3.

**Dataset Distillation: Classification** Finally in the dataset distillation task, we consider the problem of selecting prototypes to be used for few-shot classification. Here, we adopt Prototypical Networks (Snell et al., 2017) and apply the subset selection model to the task of selecting representative prototypes from each class to be used for classifying new instances. By learning to select the prototypes, we can remove outliers that would otherwise change the class prediction boundaries in the classification task. The complete graphical model for this task is given in Figure 3d where again $D_s$ corresponds to the selected prototypes and $x_*$ and $y_*$ correspond to query and class label respectively. Experimental results for this task can be found in Section 4.3.

## 4 EXPERIMENTS

In this section, we present our experimental results. Model architectures and training hyper parameters are specified in the Appendix C.

### 4.1 FEATURE SELECTION EXPERIMENTS

**Function Reconstruction - Approximation** Our first experiment is on 1D function reconstruction. Suppose that we have a function $f : [a, b] \to \mathbb{R}$. We first construct a set of data points of that function: $D = \{(x_1, y_1 = f(x_1)), (x_2, y_2 = f(x_2)), \ldots, (x_n, y_n = f(x_n))\}$ where $(x_1, x_2, \ldots, x_n)$ are uniformly distributed along the x-axis within the interval $[a, b]$. Now if we have a family of functions $(f^{(1)}, f^{(2)}, \ldots, f^{(N)})$, this will lead to a family of sets $(D^{(1)}, D^{(2)}, \ldots, D^{(N)})$. We train our model which consists of the subset selection model $p(D_s|D)$ and a task network $p(Y|X, D_s)$ (e.g. ANP), on this data set and report the reconstruction loss, which is the negative log-likelihood.

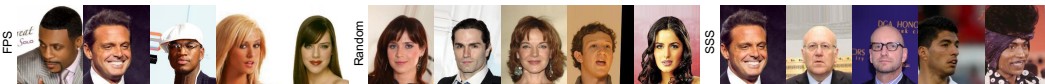

Figure 6: CelebA reconstruction samples with varying number of pixels.

Figure 7: Instance Selection Samples from a dataset of size 200.

We compare SSS with Random Select(RS) (randomly selects a subset of $D$ and uses an ANP to reconstruct the set), and Learning to Sample (LTS) (Dovrat et al., 2019) to sample k elements and uses an ANP to reconstruct the set). Figure 4a shows the performance (reconstruction loss) of our models(SSS) and the baselines.

SSS out-performs Random Select (RS), verifying that the subset selection model $p(D_s|D)$ can learn a meaningful distribution for the selected elements. Our model also out-performs the Learning to Sample (LTS) baseline.

Through the visualization of the selected points in Figure 5, we can see that out model tends to pick out more points (presented as red dots) in the drifting parts of the curve, which is reasonable since these parts are harder to reconstruct. The other two baselines sometimes fails to do that, which leads to inaccurate reconstructions.

**Image Reconstruction** Given an image we learn to select a core subset of pixels that best reconstructs the original image. Here, $x$ is 2-dimensional and $y$ is 3-dimensional for RGB images. An ANP is then used to reconstruct the remaining pixels from a set of context elements (selected subset in our case). We conduct this experiment on the CelebA dataset (Liu et al., 2018). Figure 4b shows that our model significantly outperforms ANP with RS (as in the original ANP paper) and the LTS baseline. Figure 6 shows the reconstruction samples of our model which are visually better than the reconstruction of the baselines for the same number of pixels.

## 4.2 CLASSIFICATION/REGRESSION

In this subsection, we validate our model on the prediction task. The goal is to learn to select a subset for a target task such as classification or regression. We again use the CelebA dataset, but this time the selected pixels are used to give predictions for 40 attributes of a celebrity's face (in a multi-task learning setting). For our proposed model, only the selected pixels are used for prediction (other pixels' values are set to zeros). Table 1 shows that using only 500 pixels ($\sim$1.3% of total pixels in an image), we can achieve a mean AUC of 0.9093 (99.3% of the accuracy obtained with the full image). Figure 4c shows the classification performance (in terms of mean AUC) versus the number of pixels selected. The AUC with selected pixels learned from our SSS is significantly higher than that of the random pixels baseline, showing the effectiveness of our subset selection method. We also include another baseline, namely SSS(rec). This is our stochastic subset selection model trained for reconstruction, but then later used for classification. Our model outperforms this variant, showing the effectiveness of training with the target task. Note that LTS cannot be applied to this experimental setup because during training, the generated virtual points cannot be converted back to an image in matrix form (due to the virtual coordinate), thus we cannot train the LTS model with CNN-based classification on the target task.

**Ablation Study** Since our method is stochastic, the predictive distribution can be written as $\mathbb{E}_{p(D_s|D)}[p_\theta(y_D|D_s)]$, and we can use Monte Carlo sampling to get the prediction in practice. However, throughout the experiment section, we only reported the result with one sampled subset, since it gives the best reduction in memory and computational cost. This can be seen as MC sampling with one sample. We compare it against another variant: SSS(MC) with MC sampling (5 samples). It should be noted that by doing MC sampling with 5 samples, the computational cost (inference) is increased by 5 times, and the memory requirement can be increased by up to 5 times too. Table 1 shows that our model achieves comparable performance with that variant, thus justifying that it can achieve good performance for target tasks, while reducing memory and computation requirement.

### 4.3 DATASET DISTILLATION

**Instance Selection** We present results on the instance selection task applied to a whole dataset. In this task, we use the CelebA dataset since it has an imbalance both in terms of gender and race. A dataset is constructed by sampling 200 random images from the full dataset. In this experiment, we seek to select only a few(5-30) representative images from these generated datasets. On this task, our subset selection module is trained via the procedure detailed in Section 3.6 on instance selection. To evaluate the effectiveness of the SSS model, we evaluate the model in terms of the diversity in the selected subset using the Fréchet Inception Distance(FID Score) (Heusel et al., 2017) which measures the similarity and diversity between two datasets. We compare our model with the model that randomly samples instances from the full dataset. Additionally, we compare our method with the Farthest Point Sampling(FPS) algorithm which selects $k$ points from a given set by computing distances on a metric space between all elements and selecting those elements that are furthest from each other. FPS in general seeks to obtain a wide coverage over a given set and hence is a suitable baseline. The results of this experiment is presented in Table 4 where our selection method achieves a lower FID score compared to FPS and Random Sampling. Additionally, given that the dataset is highly imbalanced, FPS performs worst since by selecting the furthest elements in the given set it cannot capture the true representation of the whole dataset even when compared with Random Sampling. Also for small sample selection, our method outperforms FPS and Random Sampling significantly since our method is able to model the interactions within the full dataset and hence can select the most representative subset.

**Classification** We use the *mini*ImageNet dataset (Vinyals et al., 2016) and go from a 20 shot classification task to one of 1,2,5 or 10 shot classification task. We again compare with Random Sampling and FPS and apply them together with SSS for the reduction in shot. The results for this experiment is shown in Table 3, where it can be observed that SSS can learn to select more representative prototypes compared to the other methods especially in the few-shot problems where the choice of prototypes matters more. All models were trained for 300 epochs and the best model was picked using a validation set.

## 5 CONCLUSION

In this paper, we have proposed a stochastic subset selection method to reduce the size of an arbitrary set while preserving performance on a target task. Our selection method utilizes a Bernoulli mask to perform candidate selection, and a stack of Categorical distributions to iteratively select a core subset from the candidate set. As a result, the selection process does take the dependencies of the set's members into account. Hence, it can select a compact set that avoids samples with redundant information. By using the compact subset in place of the original set for a target task, we can save memory, communication and computational cost. We hope that this can facilitate the use of machine learning algorithm in resource-limited systems such as mobile and embedded devices.

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

## A  APPENDIX

**Organization**   This supplementary file is organized as follows. We provide the full pseudo-code for the Greedy Training Algorithm.We then show some visualization of our method for feature selection (in both 1D function and CelebA dataset) and report the results with multiple runs of the instance selection experiment, as well as its visualization. Qualitative results for Instance Selection as applied to the few-shot classification task are provided together with model specifications.

### A.1   GREEDY TRAINING ALGORITHM

Algorithm 2 shows our greedy training algorithm with stochastic gradient descent. The idea of the greedy training algorithm is to train the auto-regressive model to select the best next $q$ elements from the candidate set to minimize the target loss on the selected samples. By doing this, we do not have to run the auto-regressive model $k/q$ time during training, thus reducing the computational cost.

---

**Algorithm 2** Greedy Training Algorithm

---

| | |
|---|---|
| **Input** | $k$(max subset size) |
| | $q$(# elements selected at each iteration) |
| | $p(D)$ (distribution of sets) |
| | $\alpha$ (learning rate) |
| | a target task with loss function $\ell(\cdot, \cdot)$ |
| **Output** | trained model with converged $\theta$ and $\phi$ |

1: $\theta, \phi \leftarrow$ initialization
2: **while** not converged **do**
3:     Sample a minibatch with $m$ sets $D^{(1)}, D^{(2)}, \ldots, D^{(m)}$ from $p(D)$
4:     $D_c^{(j)} \sim p(D_c^{(j)}|D^{(j)})$ for $j = 1 \ldots m$
5:     $i \sim$ random sample from $(0, \ldots, k - q)$
6:     $I^{(j)} \sim$ random $i$-element subset of $D_c^{(j)}$ for $j = 1 \ldots m$
7:     $Q^{(j)} \sim$ select a $q$-element subset from $D_c^{(j)} \setminus I^{(j)}$ (with the auto-regressive model)
8:     $\theta \leftarrow \theta - \alpha \nabla_\theta \frac{1}{m} \sum_{j=1}^m \ell(\cdot, I^{(j)} \cup Q^{(j)}), \phi \leftarrow \phi - \alpha \nabla_\phi \frac{1}{m} \sum_{j=1}^m \ell(\cdot, I^{(j)} \cup Q^{(j)})$

---

## B  INSTANCE SELECTION SAMPLES

In this section, we show more examples of our 1D and CelebA experiments on how the models select the set elements for the target task.

### B.0.1   1D FUNCTION - RECONSTRUCTION

Figure 8 shows the reconstruction samples of our model on the 1D function dataset, which is objectively better than that of Learning to Sample (LTS) or Random Subset (RS). Since RS selects the set elements randomly, it can leave out important part of the 1D curve leading to wrong reconstructions. LTS also selects insufficient amount of set elements in some parts of the curves, resulting in suboptimal reconstructions.

### B.1   CELEBA

Figure 9 shows the selected pixels of our model for both the classification and reconstruction task. For the attribute classification task, the model tends to select pixels mainly from the face, since the task is to classify characteristics of the person. For reconstruction, the selected pixels are more evenly distributed, since the background also contributes significantly to the reconstruction loss.

### B.2   DATASET DISTILLATION: INSTANCE SELECTION

In Table 4, we represent the full results for the Instance Selection model on the CelebA dataset. For these experiments, we construct a set by randomly sampling 200 face images from the full dataset. To evaluate the model, we create multiple such datasets and run the baselines(Random Sampling

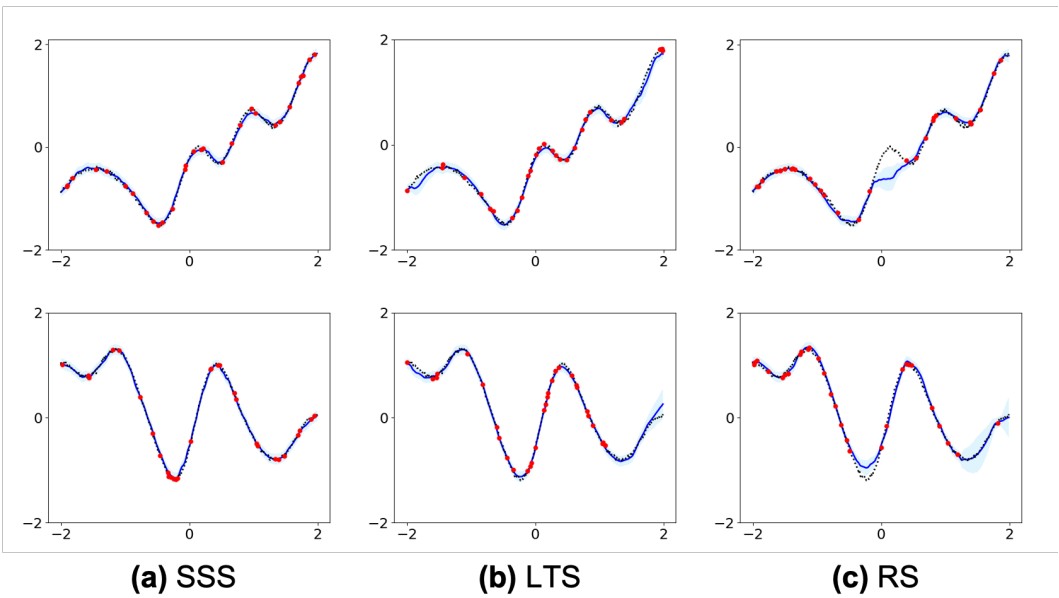

**(a)** SSS        **(b)** LTS        **(c)** RS

Figure 8: Reconstruction samples of 1D functions with different selection methods

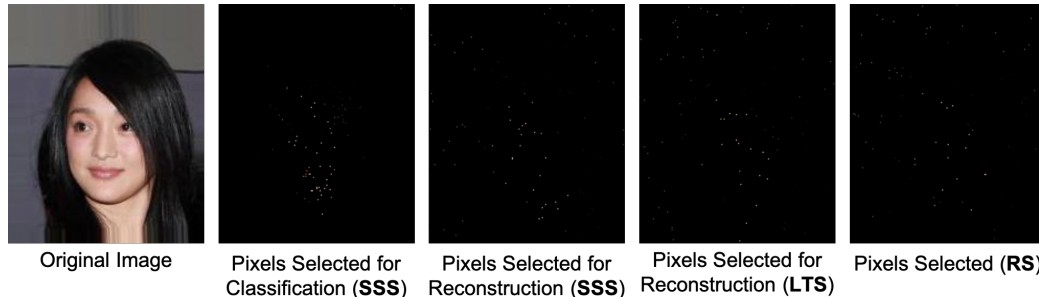

Original Image    Pixels Selected for Classification (**SSS**)    Pixels Selected for Reconstruction (**SSS**)    Pixels Selected for Reconstruction (**LTS**)    Pixels Selected (**RS**)

Figure 9: Selected pixels for different tasks on CelebA.

and FPS) and SSS on the same datasets. The FID metric is then computed on the instances and averaged on all the randomly constructed datasets. For FPS, we use the open-source implementation in https://github.com/rusty1s/pytorch_cluster. Further, we provide qualitative results on a single dataset in Figure 10 where we show how our model picks 5 instances from the full set of 200 images face images.

Table 4: . FID Score for varying Instance Selection

| #Instances | 2 | 5 | 10 | 15 | 20 | 30 |
|---|---|---|---|---|---|---|
| FPS | $6.5014 \pm 4.3502$ | $4.5098 \pm 2.3809$ | $3.0746 \pm 1.0979$ | $2.7458 \pm 0.6201$ | $2.7118 \pm 1.0410$ | $2.2943 \pm 0.8010$ |
| Random | $3.7309 \pm 1.1690$ | $1.1575 \pm 0.6532$ | $0.8970 \pm 0.4867$ | $0.3843 \pm 0.2171$ | $0.3877 \pm 0.1906$ | $0.1980 \pm 0.1080$ |
| SSS | $\mathbf{2.5307 \pm 1.3583}$ | $\mathbf{1.0186 \pm 0.1982}$ | $\mathbf{0.5922 \pm 0.3181}$ | $\mathbf{0.3331 \pm 0.1169}$ | $\mathbf{0.2381 \pm 0.1153}$ | $\mathbf{0.1679 \pm 0.0807}$ |

### B.3 DATASET DISTILLATION: CLASSIFICATION

In Figure 11 we provide visualizations for the instance selection problem as applied to the few-shot classification task. Here, we go from a 20-shot to a 1-shot classification problem where the prototype is selected from the support using SSS.

## C MODEL SPECIFICATIONS

SSS consists of $r(D)$, $\rho(d_i, r(D))$ and $f(d, D_c, D_s^{(i-1)})$. We describe the models in this section.

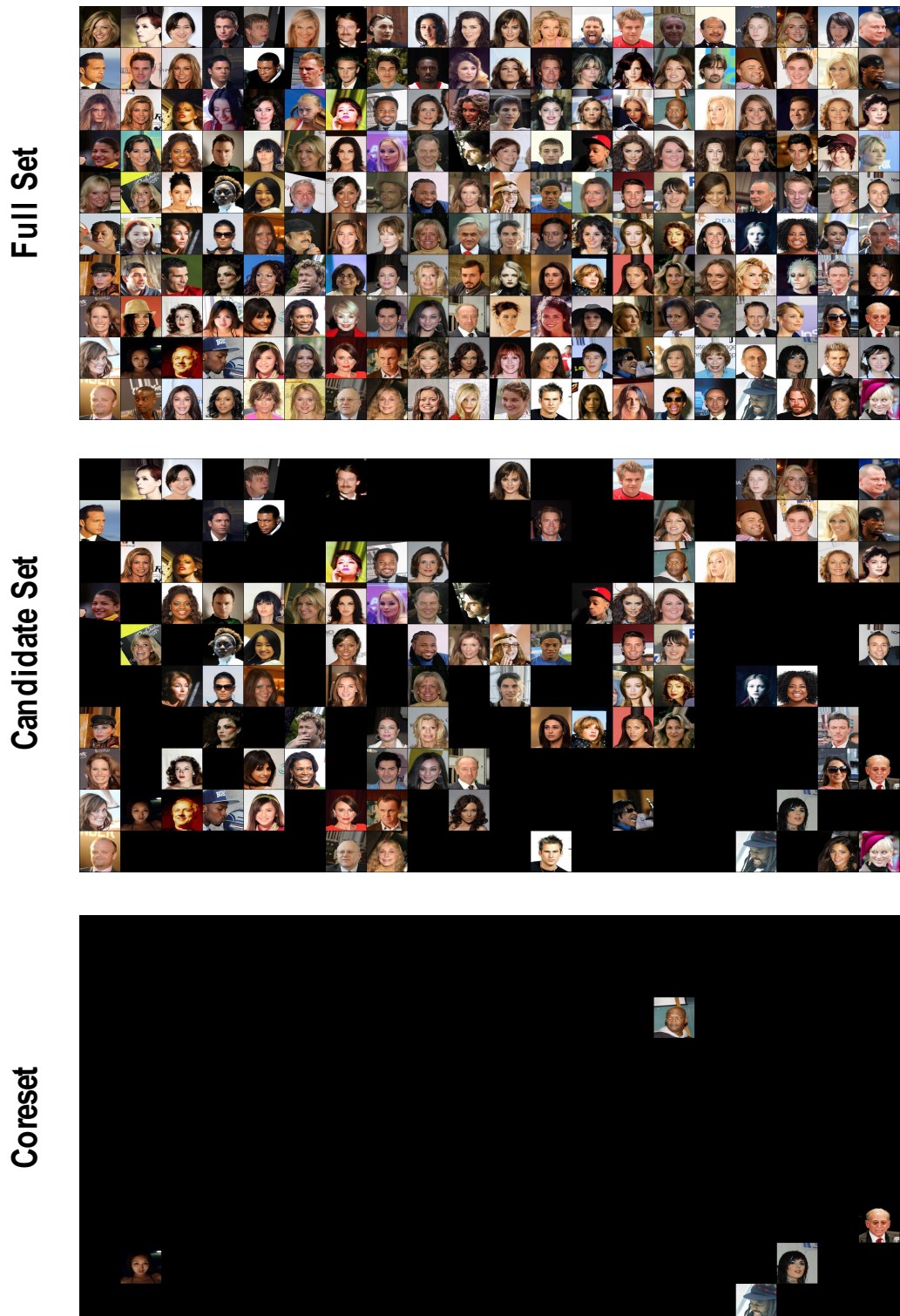

Figure 10: Visualization of a set with 200 images for instance selection. The two stage selection method in SSS is visualized as Candidate Set and SSS. A coreset of size 5 is visualized.

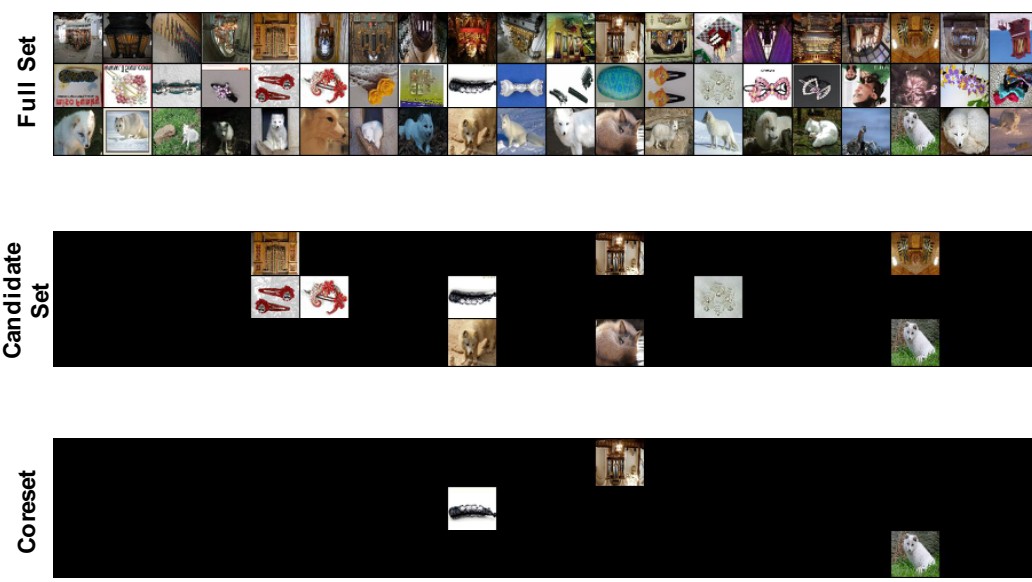

Figure 11: Sample visualization of prototype selection for the *mini*Imagenet dataset on the few-shot classification task. Each row represents a *set* that corresponds to the *support* from which a prototype is selected for the few-shot classification task.

For all experiments, $r(D)$ is implemented as DeepSets. This means that we take the mean of all the samples in a set to obtain the set representation.

$\rho(d_i, r(D))$ is implemented as a neural network with the following specifications: there are 3 Linear layers each followed by ReLU activation. Also, all inputs are projected into feature space using 3 Linear layers, each followed by ReLU activation.

In the set classification task, $f(d, D_c, D_s^{(i-1)})$ is implemented as a Set Transformer network. All other experiments use DeepSets.

