# OpenReview forum: "Stochastic Subset Selection for Efficient Training and Inference of Neural Networks"
_ICLR.cc/2021/Conference — Reject_

### Official Review · AnonReviewer2 · 2020-10-23
**interesting idea and good results but hard to read and misses the big picture**

**Rating:** 6
**Confidence:** 4

**Review:**

**Final recommendation**
I support accepting this paper. While I am a bit disappointed the authors did not add in the paper the results they discussed during the rebuttal, I think the paper is interesting and clearer than at submission.

**Summary**
stochastic subset selection (SSS) is a method to learn to compress a set $D$ by selecting a subset $D_s$ such that the loss of a task performed on $D_s$ is as close as possible to the loss if the task had been performed on the original $D$. The paper show how this general method can be applied to several tasks. For example, $D$ can contain the pixels of an image and the task can be to reconstruct the original image from the subset. This corresponds to learning to compress. As another example, with the same $D$ but a label prediction task, SSS learns to dynamically select sparse features for a classification task. The other tasks considered are dataset distillation (compressing a data set) and selecting prototypes for few-shot classification. $D_s$ is constructed in two steps. A candidate set is first constructed by considering elements of $D$ independently and in a second step elements of $D_c$ are included in $D_s$ iteratively by considering other elements in $D_c$ and in $D_s$. Experiments are performed for one example of each task previously discussed and SSS is shown to outperform baselines. SSS is motivated by the need to reduce bandwidth, computation and memory footprint of deep learning in edge devices.

---
**Strong points**

1) Code is provided (I did not try to run it).

2) The proposed approach is applied to four different tasks. This convinced me of its applicability.

3) Empirical experiments seem well conducted (with some caveat, see weak point 3) and results are good.

4) The problems tackled are interesting.

**Weak points**

1) The paper is very dense, which makes it hard to follow. Related to that, I think that some parts of the paper are too concise to understand them well. The supplementary material partly compensates that.

2) While the experiments show that the proposed approach works well, I think that some experiments should compare full data pipelines. As an example, one experiment shows that using SSS to dynamically select pixels of an image and using this sparse representation as input leads to good classification accuracy. The paper implicitely suggests that this reduces inference cost if classification is performed on the device and bandwidth if inference is performed on another device. However, the cost of the pixel selection by SSS is never analyzed. I think comparing the memory footprint, the number of operations, bandwidth used and accuracy for all steps of the following pipelines would give a much clearer picture of the interest of the approach for deep learning on mobile devices:
    - classification is done directly on the device, using an optimized network.
    - pixels are randomly selected and classification is performed on the device.
    - pixels are randomly selected, transmitted and classification is performed on another device.
    - pixels are selected by SSS, which involves computing multiple embeddings, and classification is performed on the device.
    - pixels are selected by SSS, transmitted and classification is performed on another device.

---
**Recommendation**
Overall, I vote for accepting. I am on the fence with this paper. The current results are good and I think the proposed approach has some potential and will be interesting for the community. On the other hand the paper is hard to read and the experiments, while good to show the strength of the approach, do not cover usefullness for the mobile use case. I think the former can be adressed in time and that the latter, while important, does not prevent the paper to be published.

---
**Details**

I like that each task is first formally described and then evaluated in a specific case with the experiments. However I only understood most tasks when reading the experiment sections. I would suggest to either provide a short illustration of an actual instance of the task when describing the task formally, to directly provide the experimental result after describing the task or to direct the reader to the relevant experimental section(s) for illustration.

The paper states that model descriptions are available in appendix A, but I could not find them.

For the *Set Classification/Prediction* task, I would suggest clarifying that for this task a set typically contains the features of a single example, so $D=x$. I only understood this when reading the experimental section.

The *Dataset Distillation: Instance Selection* task was not clear to me. Here are a few points that puzzled me.
- The text suggest the proposed approach is applied to each $D_i$, but figure 3(c) has no $D_i$ and $i$ seems to be used as an index of $x$ rather than $D$. Furthermore, in figure 3(c) a single D_s is constructed whereas I was expecting one per $D_i$.
- The paper states that *It is important to note that $c$ is computed using only $D_i$ for every element in $D$*. However $c$ also depends on *a query $\alpha$ which is computed using $d_i$*, which, if I understood correctly, is the element of $D$ and can be outside $D_i$. So it seems that $c$ is not computed using only $D_i$.

For the *Dataset Distillation: Classification* task, in figure 3d, the labels are not observed. How can the model learn?

1) For this task, if the label is observed, how is this different from the first task?

2) For the Image Reconstruction experiment, why is $x$ 2 dimensional? Are the inputs black and white images?

I think that the image reconstruction experiment should include a comparison to compression algorithms. I agree with the paper that the proposed approach is applicable to many other types of data than images, but it would be very interesting to see how the approach compares to expert defined methods. Furthermore, implementation in relation to mobile devices is the motivation of this work and as far as I know classical compression algorithms are still used on mobile devices.

---
**Questions**

Questions 1 and 2 in *details* above.

I would also be interested to hear authors' opinions on weak point 2 and a comparison to compression.

For classification, pixels not selected are set to 0. Wouldn't the inference cost be the same when using convolutional networks?

---
**Minor details**

For the *set reconstruction* task, would it also be interesting to learn a model on $D_s$ and use it to predict $Y$ from $X$?


*typos*
foro

---

> ### Author Response · Authors · 2020-11-18
> **Response to R2**
>
> Thank you for your constructive feedback and careful review of the paper. Below, we provide answers to the queries made.
>
> ---
> **1.1 The paper states that model descriptions are available in appendix A, but I could not find them.** Model descriptions have been updated in Appendix C.
>
> **1.2 I would suggest clarifying that for this task a set typically contains the features of a single example, so $D = x$.** Thank you for this suggestion. We have added references to each task to direct the reader to the relevant experimental section corresponding to the task.
>
> ---
> **2.1 Correction of $D$ to $D_i$ in Figure 3(c)** In Figure 3(c) $D$ is supposed to be $D_i$ where $D_i \in D$. Since $D_i$ is itself a dataset(for instance collection of images) $i$ is not used as an index of $x$. $i$ is used to index datasets in $D$. For each $D_i$, elements in $D_i$ are represented as $d$(Section 3.6 Instance Selection). With this correction, it is then clear that a single $D_s$ is constructed for each $D_i$. This correction has been made in the updated version of the paper and thank you for catching this notation error.
>
> **2.2 Computation of $c$ using $D_i$.** Following response 2.1, $c$ is computed using only $D_i$ and $d_i \in D_i$. In this case $d_i$ is an arbitrary element in $D_i$ and hence any $d_i$ is a member of the corresponding $D_i$ and not $D$.
>
> ---
> **3.1 For the *Dataset Distillation: Classification* task, in figure 3d, the labels are not observed. How can the model learn?** The model can learn because $D_s$ is fed to the classification layer in the Prototypical Network. Hence the construction of $D_s$ is dictated by the classification loss.
>
> **3.2 For this task, if the label is observed, how is it different from the first task?** The difference will be in the type of elements in the  set $D$. In the first task, elements of $D$ are the pixels in an image, while in this task, the elements of $D$ are images.
>
> **3.3 For the Image Reconstruction experiment, why is $x$ 2-dimensional? Are the inputs black and white images?** $x$ is 2-dimensional because it corresponds to pixel coordinates. This is because this experiment uses models from the Neural Processes Family(Conditional Neural Processes, Attentive Neural Processes(model used for experiments)) and these models take as input a tuple ($x,y$) where context points($y$) are the RGB values, and the target points $x$ are the coordinates of the corresponding pixels in $y$.
>
> ---
> **4.1 Comparison to Classical Compression Algorithms** We shall explore the comparison to classical compression algorithms such as JPEG for the classification task as well as the bandwidth involved.
>
> **4.2 Weak Point 2** The suggested analysis are quite useful to gain a full picture of the full data pipelines.
>   - **Pixels are randomly selected and classification is performed on the device. Pixels are randomly selected, transmitted and classification is performed on another device.** In this setting, the only difference will be the cost of transmission of the selected pixels.
>   - **Pixels are selected by SSS, which involves computing multiple embedding, and classification is performed on the device. Pixels are selected by SSS, transmitted and classification is performed on another device.** Here also, the main cost lies in the transmission cost(same as randomly selected for the same set size) plus the cost of running SSS.
>   - **Classification is done directly on the device, using an optimized network.** This also corresponds to running the model on the full input.
>   In Table 1, we provide the memory footprint for both the full model, random select and multiple version of SSS. The only outstanding cost is that of running SSS and we will analyze this and provide the results during the rebuttal period. Also, when it comes to transferring the selected pixels over a communication channel, for fixed $|D_s|$, there is no difference between the transmission cost between random selection and SSS. The main differences comes in the accuracy obtained using the selected pixels as demonstrated in Table 1.
>
> **4.3 For classification, pixels not selected are set to 0. Wouldn't the inference cost be the same when using convolutional networks?** For convolutional networks, a custom implementation of the convolution kernel will be required to take advantage of sparse computation and sparse inputs. However if we compare the situation where the selected pixels are transferred over a channel, the usage of SSS will still be useful to reduce communication cost.
>
> **4.4 For the *set reconstruction* task, would it also be interesting to learn a model on $D_s$ and use it to predict $Y$ from $X?** This would indeed
> be interesting. Our intuition is that such a model will correspond heavily with the graphical model in Figure 3(b).
>
> ---
> We provide another baseline for Instance Selection in response 1 to Reviewer 3.
>
> **References**
> [1] Kim, Hyunjik, et al. "Attentive neural processes." arXiv preprint arXiv:1901.05761 (2019).

---

> > ### Comment · AnonReviewer2 · 2020-11-20
> > **Thank you for the clarification**
> >
> > Your answers clarified a lot of points, thank you.
> >
> > I am looking forward to the results on the cost of running SSS.

---

> > > ### Author Response · Authors · 2020-11-20
> > > **On the cost of running SSS**
> > >
> > > Firstly, thank you for your response and finding our response useful in clarifying your queries.  Before providing the cost, we would like to mention that the inference complexity of SSS is given in Section 3.4 as $O(n) + O(k^2 d/q)$ where $n,d,k$ correspond to $|D|$, $|D_c|$ and $|D_s|$ respectively. We provide the cost of running SSS below.
> > >
> > > **Setting.** We report the cost of running SSS for the results in Table 1 which has the largest set size(38804 pixels). As in the experiments, we select 500 pixels in total with $q$ set to 20(thus we select 20 pixels at once as explained in Section 3.4). We measure the FLOPS and memory requirements for the parameters of SSS using [1].
> > > **Results.** For this experiments, the computational complexity of SSS is **8.38 GMac(40% of the FLOPS for the full model which has 20 GMac)** with **217.09k(compared to 958.85k for the full model)** for the number of parameters.
> > >
> > > Again, thank you for your response.
> > >
> > > **References**
> > > [1] https://github.com/sovrasov/flops-counter.pytorch

---

> > > > ### Comment · AnonReviewer2 · 2020-11-24
> > > > **so cost of running SSS is slightly less than half the cost of classification?**
> > > >
> > > > Thanks for the clarification. So, as I understand it, running SSS costs approximately the same as 40% of the cost of the classification of the full image, plus a negligible classification cost due to the much reduced number of pixels. Is that correct?
> > > >
> > > > It seems to be useful for classification on the device. I think this mostly address weak point 2 on my side. Thank you.

---

> > > > > ### Author Response · Authors · 2020-11-24
> > > > > **Yes thats correct.**
> > > > >
> > > > > Yes that is correct. SSS requires approximately 40% of the full models computational cost together with the much smaller reduced inference cost due to pixel selection.
> > > > >
> > > > > Thank you for your responses and engagement during the rebuttal period.

---

### Official Review · AnonReviewer4 · 2020-10-29
**An interesting method for subset sampling, but clarifications and better baselines are necessary**

**Rating:** 6
**Confidence:** 4

**Review:**

This work introduces a method to select instances from any set (stochastic subset selection, or SSS). The experiments demonstrate a diverse set of use-cases, including feature selection and core-set selection. The proposed approach is a two-stage method involving candidate selection (learning a function $\rho$ to determine a Bernoulli probability for each input) and AutoRegressive subset selection (learning a function $f$ to generate probabilities for sampling elements from a reduced set); both stages use the Concrete distribution to ensure differentiability.

The paper includes a broad set of experiments, but I have some questions and concerns regarding the baselines and the proposed SSS method.

1. Using the Concrete distribution enables differentiability, but Algorithm 1 seems to implement the candidate selection and AutoRegressive Subset selection steps in ways that eliminate differentiability. Line 4 assigns elements to $D_c$ based on $z_i$, but when is $z_i$ differentiated? Similarly, $Q$ is selected based on sampling using the $p_i$ parameters (line 13), but at what point are the samples differentiated after assigning elements to $Q$? The method apparently works so I'm sure there's a good answer, but it is not well explained.

2. Some aspects of SSS seem arbitrary, and the authors could make a more compelling case for this particular approach with a more serious ablation study. A couple ideas: (i) Running SSS just using the first stage (candidate selection). (ii) Running SSS using the second stage directly (if it's not too slow). (iii) Removing aspects of the models, such as removing $r(D)$ as an input to $\rho$, or $D_C$ as an input to $f$.

3. As a further baseline for SSS, could the authors simply have implemented an algorithm that learns separate Bernoulli probabilities for each example (without the model $\rho$)? This seems far simpler than SSS and it would be interesting to see how it compares.

4. When attempting to constrain the size of $D_C$, why use a KL divergence penalty and not simply penalize each $z_i$'s probability, or the sampled $z_i$'s themselves? The latter could be viewed as a version of an $\ell_0$ penalty on the subset size $|D_C|$, and both would remove the need to choose a hyperparameter for $r(Z)$.

5. Obtaining $q$ examples by independently sampling from Bernoulli distributions with probabilities given by $q \times p_j$ is odd. You can't guarantee that you get $q$ samples, and you can't even guarantee that $q \times p_j$ is a valid probability. There must be a better solution to this problem; [1, 2] both use clever sampling tricks with the Concrete distribution, perhaps they suggest a better way.

6. The paper is missing a couple citations and comparisons when it comes to feature selection using the Concrete distribution. Concrete Autoencoders (CAEs) [1] are a method for performing differentiable feature selection using a similar trick with the Concrete distribution; while CAEs are used for global feature selection (the same features for every instance), Learning to Explain (L2X) [2] is a method for instance-wise feature selection that also uses the Concrete distribution. Both should be mentioned, and L2X could be used a comparison in certain experiments.

The motivation around reducing communication overhead and allowing for faster training is interesting, but these claims could be more effectively grounded in the experiments performed in the paper. When accounting for the overhead involved in running SSS, is this method faster than conventional training, and does it require less communication overhead? The clearest application from these experiments is feature selection, but the comparisons with baselines are not extensive.

Overall, the method seems promising but I hope the authors can clarify several points and offer more competitive baselines.

[1] Abid et al. "Concrete Autoencoders for Differentiable Feature Selection and Reconstruction" (2019)

[2] Chen et al. "Learning to Explain: An Information-Theoretic Perspective on Model Interpretation" (2018)

---

> ### Author Response · Authors · 2020-11-17
> **Response To R4**
>
> Thank you for your constructive feedback. Below we provide responses to raised issues and clarifications.
>
> ---
> **1. On differentiability of $Z$ and samples assigned to Q.** As the Reviewer rightly suspects, the whole model is end-to-end differentiable. During training, assignment to $D_c$(Line 4) is performed by maintaining a differentiable mask $Z :=$ {$z_i$}$_{i=1}^{n}$ where $z_i \in $ {$0,1$}. Maintaining this mask during training ensures differentiability through the selected elements. We use the same methodology to ensure that samples in $Q$ are differentiated. In summary, in the training phase, selection for $D_c$ and $Q$ is performed through soft selection(by keeping a mask for both $D_c$ and $Q$).
>
> ---
> **2 and 3. Some aspects of SSS seem arbitrary, and the authors could make a more compelling case for this particular approach with a more serious ablation study.** We will work on the requested ablation study and provide the results during the period of the rebuttal discourse.
>
> ---
> **4. When attempting to constrain the size of $D_C$, why use a KL divergence penalty and not simply penalize each $z_i$'s probability, or the sampled $z_i$'s themselves?** In this case, a penalized loss such as the $l_0$ penalty could also be used for constraining the size of $D_C$. However doing so will require us to also chose to what extend to enforce this penalty and to fix $|D_C|$ for all sets $D$ and hence we opt to use the KL divergence instead.
>
> ---
> **5. Obtaining $q$ examples by independently sampling from Bernoulli distributions with probabilities given by $q \times p_j$ is odd. You can't guarantee that you get  $q$ samples, and you can't even guarantee that  $q \times p_j$ is a valid probability.** In practice we find that $q \times p_j$ is quite close to the original probability distribution and the size of the samples selected is quite close to $q$. The suggested papers provide a more principled way and we will further explore this and we are grateful for the pointer to these sources.
>
> ---
> **6. The paper is missing a couple citations and comparisons when it comes to feature selection using the Concrete distribution.** We have added the suggested related works to our citations. Since L2X is also used for instance-wise feature selection, we shall explore it on the tasks we have presented.
>
> ---
> We provide another baseline for Instance Selection in response 1 to Reviewer 3.

---

### Official Review · AnonReviewer1 · 2020-10-30
**The methodology makes sense, but the experimental results do not support the claims well**

**Rating:** 6
**Confidence:** 3

**Review:**

This paper proposes a stochastic subset selection method for reducing the storage / transmission cost of datasets. The proposes method minimizes the expected loss over selected datasets. The data selection algorithm consists a candidate selection stage and an autoregressive selection stage, parameterized with neural networks, and are trainable by gradient methods. The authors formulate and tested their approach on four tasks. The problem formulation and methodology are technically sound. The proposed method also seems to be more general than competing methods, such as coreset.

However, I think the experimental results do not support the paper well.
1. In Table 1 and Table 2, SSS doesn't seem to have big advantage over RS. From the selected images (e.g., Fig. 10 and 11), I also don't gain an intuition on what does SSS actually select, and how do the selected instances differ from random subsampling.
2. The paper claims to address the dataset redundancy. From to my understanding, that is, removing similar instances. However, this claim is not explicitly supported by the experiments.

---

> ### Author Response · Authors · 2020-11-13
> **Response To R1**
>
> Thank you for your constructive feedback. We address the issues raised below:
>
> **(1-1) In Table 1 and Table 2, SSS doesn't seem to have big advantage over RS.**
> - As we point out in Section 4.3, as the number of selected instances increases, it is expected that the gap between our model and the baselines decreases (In the extreme case, if the subset size is equal to the original set size, there will be no difference among the methods). Thus  more meaningful results are the performance of the models with **small selection sizes**, and our model significantly outperforms the  baselines in such settings as is evident in Table 1 where SSS has FID Score of $2.53$ while the baselines obtain $3.73$ and $6.50$ for RS and FPS respectively and Table 2 where SSS obtains $0.475$ while RS and FPS obtain $0.444$ and $0.432$ respectively. Additionally, Table 2 asserts that for small selection size, our model selects subsets most representative of the full set.
> ---
> **(1-2) I also don't gain an intuition on what does SSS actually select.**
> - Please see the **Figure 10 and 11 (In the Appendix)**, in which we provide qualitative examples of the samples selected by SSS. In Figure 10, this is especially evident. Since the CelebA dataset is biased towards Caucasian faces, random sampling from this dataset is guaranteed to be dominated by mostly samples from this set. However, it can be seen that SSS actually does select samples that cover the diverse modes in the entire set, from **race to gender**. Also, we point the Reviewer to Figures 5 and 8(for function reconstruction) and 9(pixel selection for both reconstruction and classification)  where SSS is shown to select subsets that best represent the full set for the given task. In Figures 5 and 8, it can be seen that RS neglects important sections of the functions and hence results in poorer reconstruction while SSS covers the whole scope of the functions. We provide further intuition on SSS in our response to Reviewer 3 in Concern 5.
>
> ---
>
> **(2) The paper claims to address the dataset redundancy. From to my understanding, that is, removing similar instances. However, this claim is not explicitly supported by the experiments.**
>
> - Please note that we are referring to removing elements with **redundant information**(thus using smaller number of elements for the same task), rather than **identical elements** (We do not target for such scenario, since for example, there will not be any identical elements for pixel selection since all pixels have different coordinates). We apologize about the confusion and have clarified this point in the revision.
>
> - The removal of elements with redundant information is best shown with the results of the **1D function reconstruction**experiments in **Figure 5**and **Figure 8 (in Appendix B)**. In this experiment, while LTS or RS samples from the input ranges where the function values have no curvatures, which provides redundant information, our SSS minimally samples from the input ranges with small or no curvature, while focusing on the input ranges whose function values have larger curvatures as they are more informative.
>
> - This redundancy reduction claim is also supported by the quantitative evaluation results in **Table 1**where we show that our SSS requires only a small number of pixels for the CelebA attributes classification task.

---

> > ### Comment · AnonReviewer1 · 2020-11-23
> > **Response**
> >
> > Thanks the authors for the response. I think the authors' response addressed my concerns. Therefore I raise my score to 6.

---

> > > ### Author Response · Authors · 2020-11-24
> > > **Thank you for your re-evaluation**
> > >
> > > Thank you very much for your constructive suggestions and we are glad that our response addresses your concerns.

---

### Official Review · AnonReviewer3 · 2020-11-01
**Stochastic subset selection method that is lacking in clarity, motivation, and connection to active learning literature**

**Rating:** 6
**Confidence:** 4

**Review:**

### Summary

The authors present a stochastic algorithm for selecting a subset of a large dataset, while trying to preserve statistics from the original dataset. The algorithm is a two-step process, first selecting a set of "candidates" based on individual features, then filtering to a final subset which may account for interactions between candidate items.

### Strengths

The paper provides empirical experimental results which suggest the superiority of the proposed method against two baselines: Learning to Sample (LTS) and random selection (RS).

Very few grammatical mistakes.

### Concerns and Issues that Need Clarification

1. A significant amount of active learning literature deals with subset selection. However, this paper fails to discuss or even cite any results from the active learning literature. It would be helpful for the authors to compare/contrast their proposed method with existing active learning results. For example, consider
   - [*Active Learning for Convolutional Neural Networks: A Core-Set Approach* by Sener and Savarese (2018)](https://openreview.net/forum?id=H1aIuk-RW)
   - [*Selection via Proxy: Efficient Data Selection for Deep Learning* by Coleman et al. (2019)](https://openreview.net/forum?id=HJg2b0VYDr)
   - [*Submodularity in Data Subset Selection and Active Learning* by Wei et al. (2015)](http://proceedings.mlr.press/v37/wei15.html)
   - [*On Statistical Bias In Active Learning: How and When to Fix It* (2020)](https://openreview.net/forum?id=JiYq3eqTKY) - this paper is a submission to ICLR 2020, so obviously there is no expectation that the authors would be familiar with this work. However, I bring this paper up because it discusses a similar topic of trying to select data points from a larger dataset that provide a reasonable estimate of a loss function on the "population" vs. a "sample."

2. The algorithm, as described, states that individual elements of a dataset $D$ are "possibly represented" as a pair $(x_i, y_i)$. What does "possibly represented" mean? Assuming that the dataset elements are given as a $(x_i, y_i)$ pair, the stochastic subset selection algorithm seems to describe instance selection. However, the paper also claims that the algorithm works for feature selection. What are $D$, $x_i$, and $y_i$ in the feature selection tasks? The paper makes an attempt at clarifying this by stating, "Here, x is 2-dimensional and y is 3-dimensional for RGB images." But this still leaves the reader clueless - why is $x$ only 2-dimensional? What are those 2 dimensions?

3. The end of the introduction claims that the proposed subset selection method "learns to sample a subset from a larger set with linear time complexity," yet Section 3.4 claims that the inference complexity is $O(n) + O(k^2 d/q)$, which is nonlinear if either $k$ or $d$ is chosen to be dependent on $n$. Could the authors please clarify?

4. The authors claim that their method uses meta-learning, but no mention of meta learning is made beyond the "Preliminaries" subsection of the "Approach" section. How is this method a example of meta-learning?

5. The paper provides experimental evidence that the proposed method outperforms other subset selection baselines, such as Learning to Select (LTS) and random selection (RS). However, the paper provides little (if any) intuition or formal justification for why the proposed method performs better. The authors need to explain the motivation behind the proposed algorithm.

6. The writing and figures lack clarity and are often difficult to follow. Some examples:
   - The context for the mathematical expression in (4) is very unclear. Is (4) an objective function? If so, is this expression supposed to be minimized or maximized? And what is the random variable $Z$ supposed to represent? And why is there a period in the middle of the expression?
   - Figure 2 is poorly explained and difficult to understand. I'm assuming that the blue vertical vector is supposed to represent $D_c$. Is that correct? And are the dark shaded boxes supposed to represent the "candidates" and the lightly shaded boxes supposed to represent the items that were not selected? As a reader, I should not have to guess what the colors mean. They should be explained clearly in the caption, or at the very least, in the paper text. Furthermore, it seems to me (again it's never made explicit) that Figure 2b is depicting the Autoregressive Subset Selection ("AutoSelect") routine. In this case, why is there a white row vector of $D = d_1, \dotsc, d_n$? In Algorithm 1, it seems as though AutoSelect only takes as input $D_s$ and $D_c$, which are subsets of $D$ and not $D$ itself.

### Original Rating and Confidence

**Rating** - 4: Ok but not good enough - rejection

**Confidence** - 4: The reviewer is confident but not absolutely certain that the evaluation is correct

### Updated Review after Author Response

Thank you for the thoughtful responses to my questions, and apologies for the delay in updating my review. In general the responses have addressed my questions, and I have updated my rating accordingly. However, there are still several issues in my mind:

1. **Connection to active learning** - The authors correctly point out that active learning does not assume access to labels, whereas the proposed method does. However, I believe the comparison against selection algorithms such as k-center greedy should be included in the paper for completeness sake, and not just left as a comment in this forum.

2. **Dataset Distillation: Instance Selection** - Upon re-reading the paper carefully, I found this subsection in Section 3.6 to be confusing. Why are there multiple $D_i$? The section seems to only talk about one $D_i$.

3. **Set vs. Tensor**: In the new "representation learning" subsection of Section 2, the paper stresses that the model learns a representation of a set, instead of an individual data point. However, many tasks (e.g., image reconstruction) do impose some sort of ordering. The paper lacks clarity about where permutation-invariance is used vs. not used.

4. **Other places that need clarification**:
  - Equation (2): presumably you threshold the output of the sigmoid function, right? Because in Algorithm 1, it seems like $z_i$ are supposed to be boolean-valued.
  - Appendix: there are several missing closing parentheses

---

> ### Author Response · Authors · 2020-11-13
> **Initial Response To R3 (1/2)**
>
> Thank you for your constructive feedback. Before addressing the raised concerns, we would like to emphasize that the model and algorithm presented in the paper is a general framework for both **feature (pixel) selection** and **instance selection**, and you may have **missed the feature selection**part. We address the issues raised below:
>
> **1. Comparison with Subset Selection for Active Learning**
> - Please note that our problem definition largely differs from the active learning problems tackled in the suggested papers. Firstly, while active learning **does not consider the label information**, we utilize label information. Secondly, our motivations are quite different. We focus on **efficiency in inference and training of non-parametric models by reducing the sizes of the inputs**, be it pixels(we select pixels directly from input images) or instances in the classification case, which differs greatly from the goal of active learning. Our compression of datasets is also not for labeling purposes as is common in active learning and this is backed by the experiments on instance selection in Section 4. We have included a discussion of the differences and similarities between our method and active learning in the Related Works section.
>
> - Although there does not exist a straightforward way of comparing active learning approaches and ours due to aforementioned differences, we find that almost all the referenced papers([2], [3]) utilize the **k-Center-Greedy algorithm**for instance selection and we provide a comparison between our method and this algorithm. Specifically, for each set $D$, we fix a single point $x_i \in D$ as the reference point and apply the k-Center-Greedy algorithm to select $q$ elements. For a given set $D$, the fixed point is maintained during training to conform with the active learning setting where this would correspond to the initial labeled set. We provide result for the *Dataset Distillation: Classification* task introduced in Section 3.6 on the *mini*ImageNet dataset.  From the table, we observe that  SSS outperforms k-Center greedy algorithm especially with larger gains for small selection sizes. The k-Center-Greedy algorithm performs poorly for small selection sizes since the model overfits to only a small subset of samples during training and hence the resulting model does not generalize.
>
>  | #Instances | 1     | 2     | 5     | 10    |
>  |------------|-------|-------|-------|-------|
>  |k-Center   | 0.290 | 0.413 | 0.570 | 0.656 |
>  | SSS        | **0.475** | **0.545** | **0.625** | **0.664** |
>
> ---
>
> **2-1. The algorithm, as described, states that individual elements of a dataset  are "possibly represented" as a pair . What does "possibly represented" mean?**
>
> - In the feature selection experiments, where we select a subset of the input pixels in the feature space for the reconstruction task, the input to the model(an Attentive Neural Process[1] in this case), is represented as a pair $(x_i, y_i)$ where $x_i$ corresponds to the target and $y_i$ corresponds to the context points. The phrase "possibly represented" is used to make room for such representations of the input data.
>
> **2-2. What are $D$, $x_i$, and $y_i$ in the feature selection tasks?**
> - In the feature selection task, $D$ represents the entire dataset such as CelebA. Then for a given image $I \in D$, we create context points $y_i$ which are the actual pixel values (RGB) and target points $x_i$ which are the coordinates of the pixels in $x_i$. This representation is necessary for the Attentive Neural Process[1] model, which we try to improve upon. You can find more details of the variables in Section 4.1.
>
> **2-3. "Here, $x$ is 2-dimensional and $y$ is 3-dimensional for RGB images." But this still leaves the reader clueless - why is $x$ only 2-dimensional? What are those 2 dimensions?**
> - Please note that we are selecting pixels from a whole image when applying our model for pixel selection. $x$ is 2-dimensional because it corresponds to the x and y coordinates of the corresponding pixel value, while $y$ is a 3-dimensional vector of the RGB values.
>
> ---
>
> **3. Section 3.4 claims that the inference complexity is $O(n) + O(k^2 d/q)$, which is nonlinear if either $k$ or $d$ is chosen to be dependent on $n$. Could the authors please clarify?**
> - Please note that $k$(subset size) and $d$(coreset size) are not dependent on $n$(size of full set). $k$ and $d$ are pre-determined and $d$ in particular is constrained by the regularization term defined in Equation 4. Thus both $k$ and $d$ are constant and the algorithm is linear in $n$.

---

> > ### Author Response · Authors · 2020-11-13
> > **Response To R3 (2/2)**
> >
> > **4. No mention of meta learning is made beyond the "Preliminaries" subsection of the "Approach" section. How is this method an example of meta-learning?**
> > - The proposed method can be cast into the meta learning setting with regards to some of the specific tasks presented in the experiments section since they involve training our model over a distribution of sets. As an example, in the 1-D function reconstruction and image reconstruction experiments, each function as well as image is considered a task(this is the problem formulation presented in [1] which we use for our experiments) and we train the resulting model to perform well on these tasks and to also generalize to unseen tasks(in this case unseen images). We allude to this relationship to meta-learning again in Section 3.1.
> >
> > ---
> >
> > **5. The paper provides little (if any) intuition or formal justification for why the proposed method performs better. The authors need to explain the motivation behind the proposed algorithm.**
> > - In addition to Section 3.2, we further provide some motivation and intuition behind the proposed algorithm. The main motivation is to reduce the volume of data that must be processed by modern neural networks. This not only reduces communication cost for data transfer, but also opens the avenue for performing inference on devices with low compute. As an example, the presented algorithm can improve the scalability of non-parametric models which requires the storage of training examples, such as Neural Process, to scale up to large-scale problems. on the main algorithm itself, as alluded in Section 3.3, the first stage ensures that we can first reduce the size of the input set(for instance 38804 pixels in a single image) to a small manageable set based on highly activating samples. By making the process stochastic, we can deal with larger set sizes more efficiently. The second stage, which is more fine-grained, then is meant to model rich relationships between the candidate set via pairwise interactions. The whole process is driven by the specific objective function as demonstrated by the 4 different tasks presented in Section 3.6.
> >
> > ---
> >
> > **6-1. The context for the mathematical expression in (4) is very unclear. Minimized or Maximized? And what is the random variable $Z$ supposed to represent? And why is there a period in the middle of the expression?**
> > - Indeed (4) is an objective function that is minimized over the loss function $l(.,D_s)$ and the regularizer for $Z$. We define the random $Z$ in Section 3.3 as the set $Z := $ { $z_i$ }$_{i=1}^{n}$ where $z_i \in $ {$0,1$}. As shown in Figure 2, $Z$ is the Bernoulli mask sampled in the candidate selection stage of the algorithm. There is a period in the middle of the expression $l(., D_s)$ because we leave room for an arbitrary loss function(as defined in Section 3.1) with the only condition that the loss function depends on $D_s$.
> >
> > **6-2. Figure 2 is poorly explained and difficult to understand.**
> > - We have provided detailed descriptions of the model illustration in the caption, in the revision based on your suggestion. The blue vertical vector indeed represents the candidate set $D_c$. Also, the dark shaded boxes and the lightly shaded boxes indeed correspond to the "candidates" and the unselected elements respectively. Figure 2b, correctly as you suspected, depicts the Autoregressive Subset Selection stage. The white row is supposed to show that we start with the full set $D$, use it to obtain $D_c$ and then AutoSelect uses only $D_c$ and the current $D_s$. $D$ itself is not used in this process. This will be made clearer in the updated paper and thank you very much for pointing this out.
> >
> > **References**
> > 1. Kim, Hyunjik, et al. "Attentive neural processes." arXiv preprint arXiv:1901.05761 (2019).
> > 2. Coleman, Cody, et al. "Selection via proxy: Efficient data selection for deep learning." arXiv preprint arXiv:1906.11829 (2019).
> > 3. Sener, Ozan, and Silvio Savarese. "Active learning for convolutional neural networks: A core-set approach." arXiv preprint arXiv:1708.00489 (2017).

---

> ### Author Response · Authors · 2020-11-23
> **Summary of Response**
>
> The Authors would like to kindly encourage the Reviewer to engage as the rebuttal window is soon coming to an end. To facilitate this, we provide a summary of our response to the Reviewer below:
>   - We provide a baseline with active learning methods, as requested, on which SSS outperforms the baseline.
>   - We clarify the data notation where $x$(2-dimensional) and $y$(3-dimensional) repesent pixel coordinates and pixel values respectively.
>   - We clarify that gain that the inference complexity of SSS is $O(n) + O(k^2 d/q)$ as $q$, $d$ and $k$ are not dependent on $n$.
>   - Finally, we provide further exposition on Figure 3 and clarify the link to meta-learning.
>
> We hope this brief summary of our response serves as an entry point for further discussion before the rebuttal deadline.
>
> Thanks, Authors.

---

> ### Author Response · Authors · 2020-11-25
> **The end of the interactive discussion phase is in less than 10 hours.**
>
> Dear reviewer,
>
> Could you please check our response below? We have clarified the points you have asked and provided experimental results against greedy k-center algorithm used for active learning. We have also uploaded a revision that faithfully reflects your comments. Please let us know if there is anything else we should clarify or provide, since the end of the interactive discussion phase is in less than 10 hours. We thank you for your efforts in reviewing our paper, as well as for the insightful and constructive comments.

---

### Author Response · Authors · 2020-11-18
**Summary of Updates to the Initial Version**

We would like to thank the Reviewers for their constructive feedback on the initial version of the paper. Below we provide a summary of changes made to initial version of the paper.

**1. Elaboration of Figure 2's caption.** Following Reviewer 3's comment on the caption of Figure 2, we have expanded the caption for easier understanding of the various components of the model.

**2. Expansion of Related Work Section.** Following Reviewers 3 and 4's suggestions, we have referenced works on both active learning and subset selections. Also, we have clearly stated the commonalities and differences between our target domain and active learning in Section 2.

**3. Correction of redundancy claim.** Corrections have been made in Section 3 and Section 5 to clarify the dataset redundancy claim pointed out by Reviewer 1. We have clarified that our model *removes elements with redundant information* in these revisions.

**4. Addition of references to each task in Section 3.6 to its corresponding experimental results in Section 4.** Based on Reviewer 2's suggestion, we have added references to each task description in Section 3.6 so that the experimental results on these tasks can be easily accessed.

**5. Model description in Appendix C.** We have added a description of the models used in Appendix C.

**6. Typos.** Typographical errors caught by Reviewer 2 have also been corrected.

---

> ### Author Response · Authors · 2020-11-19
> **Summary Continued**
>
> **7. Figure 3(c)** Figure 3(c) has been corrected to conform to the notation in Section 3.6

---

### Author Response · Authors · 2020-11-23
**The end of the discussion phase approaching**

Dear Reviewers,

Could you please go over our responses and the revision since we can have interactions with you only by this Tuesday (24th)? We have responded to your comments and faithfully reflected them in the revision, and provided additional experimental results that you have requested. We sincerely thank you for your time and efforts in reviewing our paper, and your insightful and constructive comments.

Thanks, Authors

---

### Decision · Program_Chairs · 2021-01-07
**Final Decision**

**Decision:**

Reject

**Comment:**

The paper proposed a two-stage method to select instances from a set, involving candidate selection (learning a function  to determine a Bernoulli probability for each input) and AutoRegressive subset selection (learning a function  to generate probabilities for sampling elements from a reduced set); both stages use the Concrete distribution to ensure differentiability. The experiments show the performance of the proposed method on several use-cases, including reconstruction of an image from a subset of its pixels, selecting sparse features for a classification task, and dataset distillation for few-shot classification. I read the paper and I agree with the reviewers that in its current format the paper is hard to follow. I strongly encourage the authors to add more discussion and intuition on the proposed method and extend the experiments with more baseline comparison and ablation studies in the revised version.